# Repeated Police Mental Health Act Detentions in England and Wales: Trauma and Recurrent Suicidality

**DOI:** 10.3390/ijerph16234786

**Published:** 2019-11-29

**Authors:** Claire Warrington

**Affiliations:** School of Applied Social Science, University of Brighton, Falmer BN1 9PH, UK; c.warrington2@brighton.ac.uk; Tel.: +44-1273-643-988

**Keywords:** Section 136, police Mental Health Act, repeated detention, suicide and suicide prevention, trauma, personality disorder, lived experience

## Abstract

Most police Mental Health Act (Section 136) detentions in England and Wales relate to suicide prevention. Despite attempts to reduce detention rates, numbers have risen almost continually. Although Section 136 has been subject to much academic and public policy scrutiny, the topic of individuals being detained on multiple occasions remains under-researched and thus poorly understood. A mixed methods study combined six in-depth interviews with people who had experienced numerous suicidal crises and police intervention, with detailed police and mental health records. A national police survey provided wider context. Consultants with lived experience of complex mental health problems jointly analysed interviews. Repeated detention is a nationally recognised issue. In South East England, it almost exclusively relates to suicide or self-harm and accounts for a third of all detentions. Females are detained with the highest frequencies. The qualitative accounts revealed complex histories of unresolved trauma that had catastrophically damaged interviewee’s relational foundations, rendering them disenfranchised from services and consigned to relying on police intervention in repeated suicidal crises. A model is proposed that offers a way to conceptualise the phenomenon of repeated detention, highlighting that long-term solutions to sustain change are imperative, as reactive-only responses can perpetuate crisis cycles.

## 1. Introduction

Under Section 136 of the Mental Health Act in England and Wales, a police officer, rather than a mental health professional, is empowered to remove an individual from a public place to a place of safety. The decision to detain is based upon the assessed likelihood in the officers’ opinion that the individual appears to present an immediate risk of causing harm to self or others by reason of their mental health. Once detained to a place of safety a Mental Health Act assessment determines whether the individual should be admitted to a mental health inpatient unit, either voluntarily or under a further Section of the Mental Health Act. If admission is not considered necessary, the person can be released. The number of such cases resulting in referral to community services is not formally recorded, but concerns have been raised at the number of detentions that do not end with any form of admission [1,2]. The number of people subject to detention who are already known to secondary care mental health services is also unrecorded, as is the incidence of individuals being detained on multiple occasions.

Section 136 (S136) has long been a controversial aspect of the Mental Health Act and scrutiny from government, leading bodies and academic literature has increased during the last decade [3,4,5,6,7,8]. One reason driving this ongoing focus has been the rates at which people have been detained, which appeared to have risen steeply during this time. Prior to 2015, detentions in which police custody was used as a place of safety were not consistently reported and consequently it was not possible to obtain accurate figures, but research for the Independent Police Complaints Commission in 2008 estimated that in the year 2005/6 there had been 17,417 uses of S136 [4]. The number of detentions in the year 2018/19 was reported to have risen to 33,238 [9].

Many complex factors are involved in the changes in S136 detention rates [10]. One highly pertinent issue that has been highlighted by several studies is that S136 is closely tied to suicide prevention. Estimates place between 55% and 81% of detentions as being linked to concerns about the detained person’s risk to their own life rather than to the threat of harming others [11,12]. A recent comprehensive study investigating the use of S136 in the adjacent counties of East and West Sussex revealed many important elements involved in the historically high rates of detention seen within that area of South East England [13]. In addition to confirming the high incidence of detentions related to suicide prevention, the findings indicated one issue requiring further investigation was that a small number of people had been detained multiple times. Examination of detention data for a 12-month period indicated that 142 people (13% of all detained) accounted for a third of all detentions, with two individuals being related to combined total of 44 detentions.

Local and national stakeholders, both professionals and people with lived experience engaged in the above study, confirmed that repeated detention was a significant issue that was felt to deserve further investigation. National data on repeated detention is not available as there is no requirement to measure or report on recurrence and very little research has examined this aspect of S136, despite many studies having indicated incidence of fewer people detained than the number of detentions examined [14,15,16,17]. However, two small studies have specifically focussed on the subject, both concluding that there was a strong correlation between those detained recurrently and the diagnoses of personality disorder, in the latter case, specifically emotionally unstable personality disorder [18,19].

The topic of diagnosis in mental illness has prompted much debate [20,21,22]. However, even within this context, the concept of emotionally unstable or borderline personality disorder stands out as particularly heavily contested. Despite long being a recognised diagnosis, the validity of the condition has been questioned in literature as well as by mental health professionals and some said to have the condition [23,24,25]. Personality disorder diagnoses have previously been seen as synonymous with being untreatable, but multiple treatment modalities are now evidenced [26,27,28] and arguments have been made for retaining the construct as beneficial for patient care and research purposes [29]. Current thinking reflected in ICD 11 has moved to a dimensional approach, classifying the severity of dysfunction with additional specific trait descriptors rather than categorising sub-types, which may allay some of the criticisms particularly around co-morbidity and accuracy [30]. However, of further concern are the negative attitudes recognised to persist within both professional and lay groups towards those said to have personality disorder [31,32,33,34]. Such negativity is often linked to the ‘challenging’ patterns of unpredictable behaviour that typify the diagnosis. Not unrelatedly, there is a high incidence of traumatic backgrounds amongst those so diagnosed, with over 90% of a cohort of patients diagnosed with borderline personality disorder having disclosed trauma [35]. Survivors of childhood trauma and multiple adversities are recognised as being at greatly elevated risk of self-harming and death by suicide, as well as having damaged relational capacities [36,37,38,39]. Poor and unstable interpersonal relationships and self-destructive behaviours, including self-harm and suicide attempts, are part of the diagnostic criteria for borderline or emotionally unstable personality disorder.

Those with a personality disorder diagnosis face a significantly reduced life expectancy compared to the general population [40,41]. Whilst this is not exclusively due to self-inflicted deaths, this group is at a 20-fold increased risk of suicide [42]. Overall, there were 6507 deaths by suicide in the UK in 2018, a significant rise from the previous year [43]. Worldwide, suicide remains one of the greatest causes of death, labelled ‘a global imperative’ by the World Health Organization [44]. Prior suicide attempts are one of the greatest risk factors predicting future completion of suicide [45]. It is, therefore, vital to investigate the links between repeated police Mental Health Act detention and suicidality.

This research has examined repeated Section 136 detentions to provide a framework by which the phenomenon can be better understood. The long-term impact of unresolved trauma was shown to have left a small number of women highly vulnerable to recurrent suicidal crises in which it was often the police who offered the only consistent response through detaining under Section 136. Inconsistencies in the responses of mental health services at times unwittingly exacerbated feelings of hopelessness that triggered suicide attempts. However, individuals who had been able to access trauma-informed support benefitted from reparative relational contexts and had been able to reduce the frequency and severity of suicidal feelings. Whilst the diagnosis of personality disorder does appear to be implicated in high frequency repeated detention, it was the core components of relationships impacting on trust and understanding that had the power to influence hope, thereby disrupting dangerous cycles of suicidal crises necessitating police detentions. This model indicates fruitful areas for intervention that could offer substantial benefits to individuals’ quality of life and significantly reduce the burden of frequent detentions on public services.

## 2. Materials and Methods 

### 2.1. Study Design

This mixed methods research incorporated three strands of data: police and mental health service information on repeated detentions within the South East of England provided a profile of people who are repeatedly detained within the region; a national police survey was used to indicate the extent to which the phenomenon was recognised elsewhere in England and Wales; and critically, in-depth interviews with six women who had survived recurrent suicidal crises and multiple detentions gave an account of the factors that drive repeated detention from the perspective of lived experience. The study drew on a realist research [46] in seeking to identify how the mechanisms functioning within a context produce an outcome, in this case, the circumstances that prompted repeated detention.

### 2.2. Police Survey

A short qualitative questionnaire was distributed through the mental health lead of College of Policing to each police force in England and Wales in February 2017. This approach was chosen to ensure full coverage of the jurisdictions of the UK to which Section 136 applies by circumventing the complexity of different NHS trusts, which are commissioned to provide varying services across the two countries. At the time police forces in England and Wales were preparing for the implementation of wide-ranging operational reform mandated in the Policing and Crime Act 2017. For this reason, the survey was developed in discussion with police advisors to be something that could be answered very quickly, with the aim of maximising responses rather than gaining in-depth information. The questions asked related to the area’s position on repeated detention; whether it was not considered an issue, through to specifics of any arrangements in place, and whether a particular frequency of detentions would be considered problematic. The survey was circulated with a brief description of the study and researcher details. Further information or contact was invited but not conditional of participating in this strand of the research. Responses were received from at least one force in 8 of the 10 regions into which constabularies are grouped [47]. This constituted an overall response rate of 30% (n = 13 replies).

### 2.3. South East Regional Detention Data

A second layer of detail was obtained from a dataset compiled for this study by the Business Manager of Sussex Partnership NHS Foundation Trust, who collates data on all Mental Health Act activity within the Trust for local and statutory monitoring and reporting purposes. Information on each incident in East and West Sussex of people being detained more than once between August 2014 and December 2016 was entered into a spreadsheet and each person was assigned an anonymous study signifier enabling the researcher to identify individuals within the parameters of the dataset only. In addition to a study case number, data provided detailed the age group, gender and number of detentions for each individual. Where identified in case notes, further variables were the primary reason for each detention, diagnosis and home area (East or West Sussex, no fixed abode, or out of area; the latter were split into neighbouring county or remote for anyone whose registered address was further away). Two-step cluster analysis using log-likelihood as the distance measure was then applied to produce an exploratory indication of groupings. This technique was chosen because it is suitable for suggesting patterning within large data sets containing mixed ordinal and categorical data. 

Three further counties within the South East of England also supplied information on their repeated detentions from existing local audit data, comprising the total number of detentions, number of people detained, number of people repeatedly detained and number of repeats, as well as the time frame to which that data related. This was reviewed with the same data extracted from the present study dataset, plus the 2012 detention data from the previous S136 in Sussex study [13] to give a broader regional indication of incidence. Owing to different levels of detail having been provided by these collaborating areas, this data was not combined but descriptive analysis was conducted.

### 2.4. Lived Experience Interviews

Finally, six in-depth semi-structured interviews were conducted with a self-selecting sample of individuals who had survived numerous suicide attempts and had experienced multiple Mental Health Act detentions, including S136. Interviews lasted between 45 and 90 min, with the length determined by the interviewee. 

Participants were recruited through two channels. The multiagency Sussex Mental Health Act Monitoring Board, which is chaired by Trust’s Principal Social Worker who is an Approved Mental Health Professional, and the Mental Health Lead for Sussex Police. The Board meets quarterly to review S136 detentions and Mental Health Act related local policy and practice issues. At the September 2015 meeting, the Board identified and contacted the ten individuals aged over 18 who had been detained with the highest frequencies over the preceding three years. Exclusionary factors were having a severe substance misuse issue or mental state diagnosis that in the opinion of the Board would compromise an individual’s ability to comprehend study information, to give informed consent or pose a risk to a lone worker. Individuals of no fixed abode were also excluded as study invites were sent by post. None of the exclusion criteria applied to the individuals who had been detained with the highest frequencies at the time of recruitment; therefore, letters were sent on behalf of the Board Chair advising that an independent researcher was conducting a study on repeated S136 detention. No personal details were shared with the researcher without individuals’ consent, but potential participants were sent details of the study and invited to contact the researcher. Two participants were recruited through this method. 

A further four participants were recruited through a specialist centre run by the same mental health trust in partnership with third sector agencies. The service is a non-residential facility that supports people with a diagnosis of personality disorder through a range of psychotherapeutic input and social support. Critically, its work is based on principles of a trauma-informed approach that centres on understanding attachment. This service was selected for recruitment because the study was likely to be of interest to several of the members. The researcher attended a community meeting and explained the study. A sign-up sheet was left with the receptionist and the researcher then maintained contact with the centre over the following three months, attending to discuss the study further with interested members. The four members who chose to take part in interviews had each been engaged with the service in excess of a year. 

### 2.5. Qualitative Data Analysis

Thematic analysis [48] of interviews was used to enable inductive identification of themes within the data. Interviews were audio recorded and were transcribed verbatim by the researcher soon after completion. The transcripts were then reviewed for accuracy and at the same time initial codes were highlighted using a data-driven approach given the exploratory nature of the research. This preliminary analysis was reviewed with experienced senior academic colleagues who suggested a sufficient level of data had been obtained to yield a comprehensive analysis in combination with the quantitative data. Recruitment had simultaneously reached a natural point of closure with no additional potential participants making contact. The researcher, therefore, chose not to seek further participants. Initial candidate themes were next collated from a further iterative review of the preliminary codes and related excerpts of the interviews. These themes were then reviewed in a workshop format analysis meeting with members of the same NHS Trust’s Patient and Public Involvement (PPI) Research Advisory Panel. This is a group of patients who have experience of complex mental health problems and crises and who work together to consult on related research within the Trust from the perspective of lived experience. Interviewees were assigned pseudonyms and identifying information was concealed by the researcher prior to this stage. The joint analysis took the form of a three-hour meeting that was attended by five female lived experience consultants, one of whom co-facilitated the meeting with the researcher. During the meeting an overview of the study was presented, which was followed by a discussion around the initial candidate themes with excerpts from the transcripts. The co-facilitators then posed questions to the consultants to prompt consideration of their interpretations of the data in relation to the initial themes. At the end of the meeting, the discussions were summarised, leading to revised themes being agreed upon. The entire qualitative dataset was then reviewed again by the researcher examining the extent to which the data fit the refined themes produced in the joint analysis. 

### 2.6. Ethics

As well as guiding the analysis through the above process, PPI consultants supported this research through scrutiny from initial design, pre-ethics scrutiny and advice on study documents. Favourable opinion under NHS Health Research Authority was given for the qualitative research element by NRES Committee South East Coast —Brighton & Sussex (15/LO/1219). The remaining data collection was approved by South East Coast—Surrey Research Ethics Committee (16/LO/2069). 

## 3. Findings

### 3.1. Service Perspectives on Repeated Detention

All police survey respondents recognised repeated detention occurring to some extent within their area, with some adding they felt it signalled unmet need. Only one reply stated it was not considered a significant issue as only four people had been detained on multiple occasions in that area in the previous 18 months. All respondents felt repeats necessitated a multiagency response. Rather than attempting to precisely define a threshold at which detentions should be considered problematic and trigger a review, the replies received indicated the circumstances of an individual’s detention should be scrutinised to dictate what action services may need to take to support the person away from recurrent crises.

Although the police survey did not ask for specific information on the reasons for repeated detentions, several replies indicated that multiple detentions in respondents’ forces were linked to suicide prevention and mentioned high-risk behaviour linked to certain geographic locations in which people were recognised as being detained on several occasions.

The local dataset produced for this study contained 563 multiple detentions of 155 individuals that took place during the index period of 28 months. The reason for detention was unknown in 40 incidents (7%) but suicide or deliberate self-harm was the reason for the greatest number of detentions (481). Excluding those cases with missing data this equated to 92%. Suicide was identified as a secondary factor in a further 16 (3%) incidents. Herein, detention had been based on the apparent risk posed by the detained person to others or their mental state being of greater concern than their potential risk of self-harm. Examples in this category included people carrying a weapon in busy public places with which they were threatening to harm themselves. Only 26 detentions (5%) were entirely unrelated to concerns about the detained person’s risk to self. Not only does this data provide additional strong support for the connection between S136 and suicide prevention but it further indicates that repeated detention is particularly linked to self-harm.

Most responses to the police survey identified some form of regular or ad-hoc monitoring of repeats was in place. Likewise, the audit data compiled from police and mental health services covering three other areas within the South East region of England reflected that concerns about some individuals coming to repeated police notice through S136 had led to this information already being monitored at local levels. Despite covering different time frames as well as diverse localities, this regional data, shown in Table 1, revealed strong similarities in the proportions of multiple detentions, indicating an overall average of 31% of all S136 detentions being repeats.

Analysis of the Sussex dataset specifically compiled for this study showed considerable gender differences across the number of detentions per individual. As shown in Figure 1, half of all people subject to multiple S136 were detained twice (51%) and 57% of this group were male. However, overall only 45% of those subjected to repeated detention were male and it was exclusively females who were detained with the highest frequencies.

Diagnosis information was missing for 34 people who were detained on 84 occasions, these cases were excluded from the subsequent analyses. Among the remaining individuals who were detained repeatedly, 58 people (49%) with single diagnoses of personality disorder accounted for a total of 262 detentions (55%). The diagnosis linked to the highest number of detentions was emotionally unstable personality disorder, four other subtypes of personality disorder were recorded in relation to eight people detained on 20 occasions. A further 20 individuals (17%) who were detained on 86 occasions (18%) were recorded as having multiple diagnoses that included some form of personality disorder. A broad range of other diagnoses including learning disability, depression, anxiety, post-traumatic stress disorder, psychoses and substance misuse disorders were recorded for 43 (36%) people who were related to 131 (28%) detentions, but no single diagnosis type related to more than ten detentions.

Owing to the small numbers of individuals observed with many common diagnoses, the data was divided into single diagnosis of any personality disorder; diagnosis of any personality disorder plus one or more other diagnosis and any other diagnosis. Two-step cluster analysis suggested a model containing three sub-groupings. As shown in Table 2, the greatest number of detentions (229 in total, 48%) were related to the first cluster. This group comprised 49 females, with a mean age of 27 (SD 8.21). Each person in this group had a single diagnosis recorded of either borderline or emotionally unstable personality disorder. The remaining 50 females for whom diagnosis data was recorded formed the second cluster which was related to 102 detentions. The average age of the second group was 28 (SD 9.85). Finally, the third group constituted the 29 males within the dataset for whom any diagnosis was recorded. This group represented 145 (31%) detentions and had an average age of 34 (SD 11.12).

### 3.2. Lived Experience of Repeated Suicidal Crises

Multifaceted issues including histories of trauma, often experiences of childhood abuse, marked participants’ narratives. The traumatic experiences themselves were not cited as the proximal causes of suicide attempts; rather, it was the enduring legacy of unresolved psychological impact, particularly as manifesting in relational problems. Medicalised approaches employed by mental health services were commonly experienced as disempowering and could exacerbate these difficulties and mediated participants’ suicidal feelings.

The lived experience advisors who jointly analysed the qualitative data agreed unanimously that ‘relationships’ was the most critical theme. This has been highlighted as of great importance for patients diagnosed with personality disorder as a central feature of both the development and resolution of difficulties [49]. Interviewees described how their relational contexts could determine whether feelings of hopelessness had been triggered or managed at varying times.

Many participants spoke about feeling isolated. Prior to joining the support service Diane described herself as having been “*a bit of a hermit*” saying that being overwhelmed by anxiety, she had barely left her house and rarely saw anyone other than her care co-ordinator who had visited her at home. Professional relationships have greater importance for patients with serious mental illness [50] and other participants also mentioned these as key in talking about themselves. Kate stated:


*“I see my psychotherapist once a week, he is probably the only person who understands me. But him understanding me doesn’t change my life.”*


Likewise, another interviewee, Anna said that prior to joining the support service her care co-ordinator had been*: “doing his best to help me, but he couldn’t do it all on his own.”* This she said had meant that during that time she had often felt suicide was her only option as she could not be helped. 

Another element of this theme touched on by some interviewees was personal relationships that were experienced as being supportive yet tinged with guilt. As the interview further explored her social network, Kate said *“the one person who has stuck by me”* was a friend to whom she felt she was a burden. Similarly, Heather talked about her husband being very supportive of her, but she described great guilt at how her mental illness had affected him, saying it had “*ruined*” their marriage.

Interviewees engaged with the support service benefitted from consistency in both their professional and personal relationships developed with other members. This appeared to provide a reparative environment that had fostered positive change. Beth explained that the service purposefully works to ensure that all staff know each member as an individual:


*“You know they do get to know you, and I think their sort of ethos is to get you to bond with the whole team rather than just the one person. So you do have a key worker but there’s always somebody else there …so you’ve got the whole team really not just one person.”*


Anna reiterated this, explaining how the collective approach contrasted her previous good relationship with her care co-ordinator, which although supportive and beneficial had been insufficient to move her past frequent suicide attempts:


*“they work as a team. Know you as a team. And that makes me feel a lot safer. That makes me feel listened to. I feel supported. I feel like the staff understand… the nature of this illness. How it affects us.”*


Conversely, inconsistent relationships with both family members and care workers for some participants mirrored or continued the circumstances surrounding their traumatic backgrounds. Diane said she had also been told at one time that the mental health team were not a long-term service and that her case may be closed, although she felt she needed ongoing specialist support. As a result of these experiences, she said she felt vulnerable and that each time she has a new care co-ordinator assigned to her it would take her several months to begin to believe they would not discharge her from the service. Related to this, participants mentioned the different approaches of different mental health professionals coupled with how frequently in some cases care teams were changed, meant individuals were left feeling unsettled.

In many cases relational contexts had also been implicated in the genesis of individuals’ difficulties, experiencing abuses perpetrated within previously assumed trusted relationships. Feeling disempowered or that an element of betrayal has been implicated in trauma has been strongly linked to traumagenic responses becoming embedded [51,52]. Complexities in relationships with their own children also provoked distressing feelings of guilt and failure. A uniquely harrowing statement from Kate expressed this sentiment:


*“I’ve failed at my marriage. I’ve failed at being a parent. Suicide’s just another thing I’ve failed at.”*


Relationships heavily influenced the other components within the model, enabling activation when the context was favourable, as shown in Figure 2. The mechanistic elements of trust and understanding operated in tandem. The experience of being understood and well-supported had over time enabled participants within this positive context to reconstruct their self-images. For example, Anna stated that she had come to realise that her extreme thinking and resulting patterns of self-destructive behaviour had developed as responses to the multiple traumas she had survived, reflecting on the change the support service had facilitated for her she said:


*“I was taking regular overdoses, probably monthly, because I just couldn’t manage the feelings… I couldn’t manage feeling such a failure. I couldn’t manage… the pain of living. I couldn’t see that it would ever end.”*


A sense of being understood was crucial for all participants. Yet as Kate’s words about her therapist had indicated, this of itself was insufficient to provide a sense of safety. Nonetheless, services demonstrating an understanding of the way in which their decision-making could be interpreted was of great significance. Several interviewees spoke of the impact of being rapidly discharged without explanation, for example, having received resuscitation in hospital following a suicide attempt, Anna had been discharged to the care of a Crisis team who had closed her case the following day. Recalling this, Anna said:


*“they didn’t communicate to me the reasons why they wouldn’t work with me. I felt rejected and abandoned. And like no one cared about what would happen to me.”*


She had subsequently taken another overdose, believing she could not be helped and said she had initially been furious when she had been taken back to hospital as she had been intent on ending her life. Heather had similarly been discharged from hospital being told by staff they felt she was worse than at admission. She observed that:


*“The attitude of services is ‘if you have a personality disorder diagnosis, we can’t help you’.”*


A seeming failure to consider the impact of how communication around clinical decisions could be framed spoke to lack of understanding that impeded recovery. Diane too spoke of an occasion in which she had been admitted to hospital late at night from a S136 detention but discharged by a different psychiatrist when ward rounds were conducted the next morning:


*“when I got to [name of hospital] I felt safe, and calm, and then the doctor, the next day turned round and says ‘Yeah, oh yeah I know her she’s ok, I’ll take her off Section. She can go home’.”*


Diane’s experiences also pointed to ‘trust’, which was allied to understanding. Trust emerged from the foundation of consistently stable relationships in which understanding was developed and maintained. Heather avowed her ability to trust professionals had been shattered by detrimental contexts in which she had experienced inconsistency founded on insufficient understanding:


*“I’m too frightened to ask for help for fear of rejection and then feeling even more alone. So, I won’t ask for a hospital admission… I wonder how much the mental health system has contributed to me feeling constantly suicidal.”*


The final element of the model related to whether feelings of hope were activated, resulting in suicidal feelings being managed. In the reverse, feelings of hopelessness became overwhelming and triggered individuals to act on suicidal feelings. Participants describing positive contexts in which they benefitted from multiple consistent relationships that were stable over time spoke with a distinct future orientation and referred to their futures with a sense of agency. Anna and Beth both spoke about learning to anticipate and plan for difficult situations, indicating implicit beliefs in their further recoveries in the future. Emma too spoke about plans agreed with her care team for her to engage in trauma-focussed therapy within the next few months. Beth echoed a similar sentiment, speaking assuredly about her future she referred to the support centre as: *“a service I’m hoping to move on from.”* Diane described the service as having saved her life and Anna said her self-harming as well as suicide attempts had been virtually eliminated. In stark contrast, participants who felt they did not have access to adequate support revealed ambivalence toward their ability to sustain their own lives in the long term. 

Hopelessness was recurrently produced as the culmination of the elements described, leaving individuals feeling powerless to overcome suicidal feelings. When reaching this point of crisis, feelings of being a burden to close contacts or being unable to seek support from other services, the police offered the only consistent response in intervening in suicide attempts. Kate said she found the process of being detained harrowing and that it was something: *“I wouldn’t wish on my worst enemy.”* Likewise, Heather described various strategies she had employed to attempt to avoid being detained by the police when trying to take her life. Reflecting on this she stated: 


*“The police are the only people who have to do something. They can’t leave you... So, I have really mixed feelings on 136.”*


Heather and Kate had both concluded there was little their mental health teams could offer to make a meaningful difference. For Kate, an out of area therapeutic placement had broken down, leaving her to feel she was unmanageable, and Heather had been told the form of trauma therapy most likely to benefit her was not available. She said this left her feeling that: 


*“Everything that happens is merely a sticking plaster until the next 136. [Services] know it. I know it. [so] half of me wants some help, the other half wants to be dead.”*


## 4. Discussion

This study has taken a multifactorial approach to develop an understanding of repeated detention by the police under S136 of the Mental Health Act. The research has demonstrated the phenomenon to be widely recognised throughout England and Wales, and to predominantly relate to suicide prevention and self-harm. Repeated detention accounts for close to a third of all detentions. It is a small number of females, likely diagnosed with emotionally unstable personality disorder, who are detained with the highest frequencies. For these individuals, the consequences of prior experiences of trauma are commonly exacerbated in detrimental contexts of inconsistent relationships, lack of understanding and mistrust, prompting hopelessness and recurrent suicide attempts. In the absence of adequate systemic support, the police provide a consistent response by intervening repeatedly to detain. However, this model can operate in reverse to produce beneficial outcomes. A favourable relational context that is maintained over time can enable understanding to generate trust, stimulating hope and thereby reducing the frequency and intensity of suicidal feelings.

Many of those detained more than once were males who were subject to two detentions, suggesting this group may be representative of the wider cohort of people detained under S136, for whom a period of crisis culminates in police intervention that can then prompt appropriate help-seeking and access to support services, albeit through a harrowing experience of being detained [13,53]. However, this research has confirmed that those subject to the highest numbers of police Mental Health Act detentions are almost exclusively female. This is the inverse of national detention data and of overall S136 trends in the study locality. Likewise, locally, nationally, and worldwide, more men than women die by suicide. These two factors may have contributed to hitherto obscuring the needs of the small number of females who are subject to frequent detentions.

Information was not specifically sought about the circumstances surrounding each person’s historical traumas, but these experiences were nonetheless apparent throughout interviewees’ narratives. The accounts in both this and the prior S136 in Sussex study contained multiple examples of women whose diagnosis of emotionally unstable personality disorder appeared to have been heavily based on their repeated suicide attempts and histories of abuse. Yet this diagnosis is still too frequently experienced as ‘de facto demedicalization’ [32], being an exclusionary factor from many mainstream mental health services. The entrenched nature of the relational difficulties that are frequently seen as one of the primary indications of borderline or emotionally unstable personality disorder commonly see this group of patients marked out as hard to engage and challenging to work with, often owing to the difficult emotions working with such emotional instability can provoke in staff [42]. Treatments for patients with personality disorder, especially those drawing on trauma informed approaches that seek to support relational impairment from a non-judgemental and open stance that support staff to understand the likely impact of past experiences on the individual have shown significant benefits to both patients and workers [19,54,55]. However, healthcare professionals can consider those identified as having borderline or emotionally unstable personality disorder to be less ill than other psychiatric patients and thus less deserving of empathy [56]. This study has provided further support to findings that negative healthcare attitudes can be counter-therapeutic [57]. In turn, this can contribute to increasingly desperate attempts to access support that are all too often dismissed as ‘attention-seeking behaviour.’

A sense of rejection from mental health services can be experienced as reiterating individual’s feelings of worthlessness and hopelessness. Believing there is nothing that could alleviate their recurrent distress drains the individual’s scant resources to combat suicidal urges [58]. Participants’ descriptions of suicidal feelings and prior attempts indicated the presence or absence of hope influenced the course of action when in crisis. An inability to conceptualise a future was shared by both the interviewees who did not have access to sufficient support and the accounts of their experiences prior to joining the support service by participants who were members of the centre. All too often, this sense of futility had seemingly been reinforced by interpretations of inconsistencies perpetuated by mental health services. Hopelessness inevitably features in multiple models of suicide [59,60,61]. The present study strengthens understanding of the pathways to suicide attempts, enhancing the interpretations of how attachment [62] and agency [63] are likely mediators in the development of ‘suicidal exhaustion’ [58]. These factors are of particular salience to women diagnosed with emotionally unstable personality disorder, for whom histories of childhood adversity and trauma can become the overarching context within which the inherent inconsistencies of an under-resourced mental health service can too often be re-traumatising with devastating effect.

Whilst many women diagnosed with borderline or emotionally unstable personality disorder will survive numerous suicide attempts, self-inflicted death remains a very real and significant risk that cannot be ignored. The progress support service interviewees had made in managing their mental health and no longer falling victim to uncontrollable suicidality is testament to the possibilities inherent in services adopting a trauma-informed long-term view to supporting vulnerable adults over time. It is thus imperative that rather than dismissing or excluding trauma survivors, services adopt long-term approaches to work with individuals’ strengths. This research has underscored the need for interventions that can disrupt the cycle of suicidal crises and police detention. 

It should be noted that although strength of this study is in the variety of data sources employed, the self-selecting samples of both police survey respondents and interviewees is likely to have influenced the findings. All interviewees were working-age adult females; therefore, further research would be necessary to understand whether the model applies to other cohorts. Extending the number of participants and including more detailed perspectives from police and mental health professionals would also greatly strengthen these findings. 

## 5. Conclusions

Understanding repeated S136 Mental Health Act detention through both the lens of lived experience and the perspectives of police and mental health services is critical to developing safe pathways away from the dangerous cycle of despair and crisis. This study has highlighted how a consistent relational context that remains stable over time can enable individuals to manage recurrent suicidality through repairing the fractured self-image that can result from surviving trauma and adversity. This has important implications for multi-agency police and mental health partnerships to work with those for whom frequent police intervention has become a recurrent experience.

## Figures and Tables

**Figure 1 ijerph-16-04786-f001:**
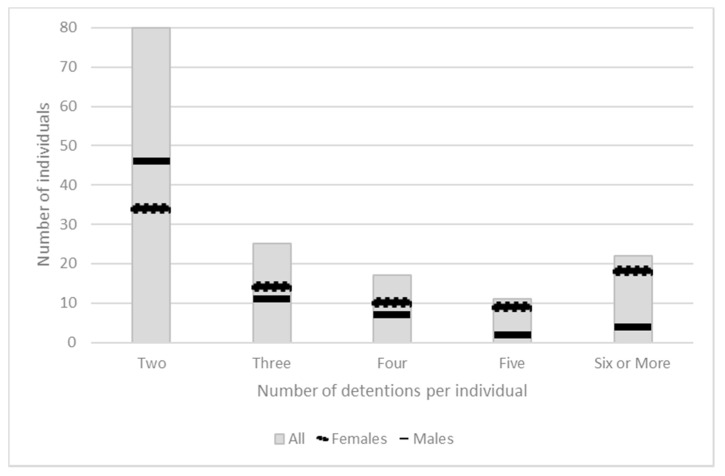
Number of multiple detentions by gender.

**Figure 2 ijerph-16-04786-f002:**
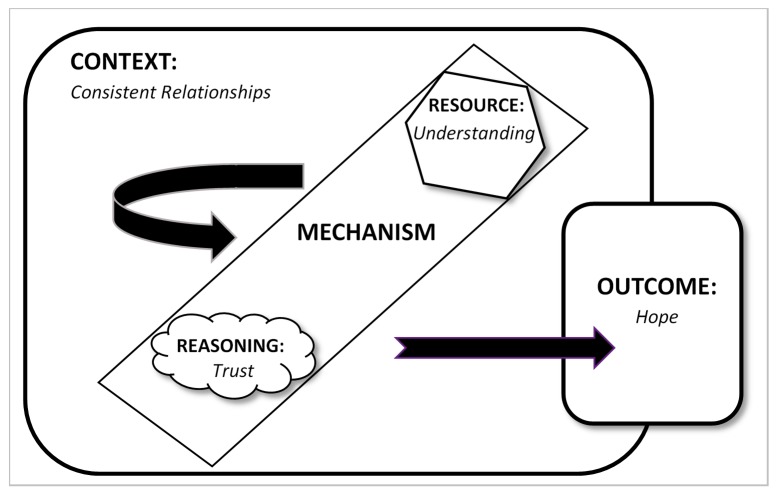
Positive relational context triggering hope by the activation of understanding and trust.

**Table 1 ijerph-16-04786-t001:** Proportion of repeated detentions in South East region.

Area	Dataset Length	Total Number of		Proportion of	
		Detentions in Dataset	People in Dataset	Repeated Detentions	People Detained Repeatedly
A *	28 months	2611	2203	22%	7%
A	12 months	1421	1142	30%	13%
B	36 months	1091	821	37%	16%
C	12 months	171	69	32%	12%
D	6 months	601	475	32%	13%
Average		1179	942	31%	12%

* Information extracted from dataset for this study. Other data derived from locally provided audit data.

**Table 2 ijerph-16-04786-t002:** Sub-groupings of those repeatedly detained.

Characteristics	Number of Associated			
	Detentions		Individuals	
Females with sole diagnoses of a personality disorder	229	(48%)	49	(38%)
Females with any other diagnoses	102	(21%)	50	(39%)
All males in dataset (all diagnoses)	145	(31%)	29	(23%)

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
