# Peer review of "Repeated Police Mental Health Act Detentions in England and Wales: Trauma and Recurrent Suicidality"

_ijerph, 2019, doi:10.3390/ijerph16234786_

Round 1

Reviewer 1 Report

This paper reports the findings of a study addressing a sensitive topic of detention in the context of mental health and suicidality. I acknowledge the importance of research in this field, and the study seemed to produce compelling findings. However, in a study like this, the research methodology is of critical importance. Regrettably, the description of the methods raises important questions that must be addressed.

The study methods comprised three aspects.

(i) a qualitative questionnaire sent to local police forces.

What was the content of this questionnaire? How was it prepared? How was it analysed? Quantitatively or qualitive?

On line 114 it is said that the questionnaire ‘yielded a representative indication’: Please clarify how the representativeness was assessed.

(ii) an anonymous dataset compiled for the study. How were the data collected? What type of data? Were standardized instruments included? What is known of the reliability and consistency of the instrument? How were data analysed?

On line 122 it is said that data from selected counties were added to the database. Did they use the same instrument to collect their data? What is the reliability compared to the data that were collected for the study? What was the process of merging the data?

(iii) semi-structured interviews with six women. What were the inclusion and exclusion criteria? How come the study selected only female participants? Surely this introduced a serious bias. Also, if detention is such a major problem, then why limit the sample to six participants?

It seems that authors conducted a thematic analysis? How was this approached?

Line 144 said that the thematic analysis involved ‘consultants’. What were their qualifications or training to be involved in a scientific analysis? How were they selected? Inclusion, exclusion criteria? What was their gender distribution?

Given the many methodological concerns I will refrain from commenting on the results and conclusions.

Good luck with a revision.

Reviewer 2 Report

The paper is well written, matherials and methods are clear and well done. Results are well expressed.

Author Response

My thanks to reviewer 2 for their time and consideration of my manuscript.

Round 2

Reviewer 1 Report

Dear Author,

Thank you for considering my comments. I have no further questions. 

Best wishes,